# Thermal Effectiveness Enhancement of Autoclaved Aerated Concrete Wall with PCM-Contained Conical Holes to Reduce the Cooling Load

**DOI:** 10.3390/ma12132170

**Published:** 2019-07-06

**Authors:** Atthakorn Thongtha, Aitthi Khongthon, Thitinun Boonsri, Chan Hoy-Yen

**Affiliations:** 1Department of Physics, Faculty of Science, Naresuan University, Phitsanulok 65000, Thailand; 2ASEAN Centre for Energy, Kota Jakarta Selatan, Daerah Khusus Ibukota Jakarta, 12950, Indonesia

**Keywords:** autoclaved aerated concrete, insulation materials, phase change material, conical hole, cooling load, energy saving

## Abstract

This work investigates and improves the thermal dynamics of autoclaved aerated concrete (AAC) wall containing phase change material (PCM). The PCM is paraffin wax loaded into conical holes drilled into the AAC. Filled AAC with three different numbers of PCM-filled holes (2, 3, and 4 conical holes, which are designated as AAC-2H, AAC-3H, and AAC-4H, respectively) as well as the unfilled original AAC were both tested under two different conditions: indoors (with controlled temperature) and outdoors (with actual weather). For the indoor experiment, a heater was used as a thermal source and set up to maintain the testing temperature at one of three levels: 40 °C, 50 °C, or 60 °C. The wall temperature was then measured on the surface with each horizontally-positioned wall as well as four different positions at various depths below the surface of the wall. It was found that AAC-4H was the optimum condition, which can produce outstandingly a time lag of approximately 27%, reduce a decrement factor of approximately 31%, and also decrease the room temperature. This reached approximately 9% when compared with that of ordinary AAC at the controlled testing temperature of 60 °C. All samples were further tested in actual weather to confirm the thermal performances of AAC-4H. Thermal effectiveness of AAC-4H was improved by extending approximately a 14.3% time lag, which reduces approximately a 4.3% decrement factor and achieving approximately 5% lower room temperature when compared with ordinary AAC.

## 1. Introduction

Electricity consumption worldwide in the building sector has grown from 26% in 1980 to 54% in 2010, and is forecasted to be 84% in 2050 [1,2]. The electricity consumption in the building sector is due mainly to the air conditioning system and mechanical ventilation to obtain indoor thermal comfort. Up to approximately 64% of the electricity consumption in South Asia is due to air conditioning because of the hot weather in the region all year round [1,2]. Building cooling systems in high solar radiation areas are high energy consumers. Therefore, research is being done on the building envelope to decrease thermal accumulation in buildings [3,4,5,6].

Currently, concrete is widely used as an important construction material in residential housing, public buildings, offices, and thermal energy storage materials. To reduce the environmental impact, and improve mechanical and thermal properties during concrete production, G.L. Golewski [7,8] used fly ash as a component of concrete to decrease fly ash landfills by 160 million tons. This usage helped reduce the environmental impact, and improve the mechanical properties of the concrete. Furthermore, the thermal properties of the concrete were improved, which enables its use as a thermal energy storage material in solar-thermal power plants [9,10,11].

In residential housing, the application of new concrete types in the building envelope, to decrease heat gain, is key to reducing energy consumption. These new materials are required to improve building designs to reduce energy consumption. One of the approaches for reducing energy consumption in buildings and improving interior thermal surroundings is the integration of phase change material (PCM) into a building or building services system, which was first applied for thermal storage in buildings in 1980 [12], to improve the efficiency of thermal storage [13,14]. Experiments that used PCMs in trombe walls [15,16], wall boards [17,18], shutters or windows [19], ceiling boards [20,21], and roofs [22,23,24] have reported reduced energy consumption.

For the efficiency of thermal storage, metal foams were incorporated into PCM to increase the potential of thermo-mechanical properties for high temperature thermal energy storage [14]. Zhou and Pang [15] enhanced the thermal behavior of a PCM in Trombe wall systems for passive solar heating by using longitudinal vortex generators. which can improve the gap air flow rate of around 28.5% and a heating rate of around 39.4% to the room when compared with the condition without vortex generators. Li et al. improved the thermal performance of the Trombe wall integrated with double layers of phase change materials that can enhance indoor thermal comfort and reduce the cooling/heating load over the whole year [16]. PCM composite board with the incorporation of shape-stabilized PCM particles in a polymer matrix applied to the thermal insulation that showed optimized performances during the summer, while exhibiting a very poor choice during the winter [17]. Furthermore, the gypsum board when incorporating 45% by weight of PCM increased heat storage capacity and decreased energy consumption [18]. Li et al. incorporated paraffin wax with added nanoparticles into double glazed windows and obtained the condition of nanoparticle concentration of 1% and nanoparticle diameter of 100 nm met minimal energy consumption for all seasons [19]. Yasin et al. made the chilled PCM ceiling model by simulating massive thermo-active building elements with PCM, which corresponded to the real scale building data [20]. Next, PCM was used in buildings with a ceiling fan-assisted ventilation system to improve indoor thermal comfort and to shift cooling/heating energy demand away from peak hours [21]. Chou et al. improved the metal-sheet roofing structure by using PCM, which can reduce heat transfer through the roof to the indoor space and maintain indoor comfort for longer [22]. Lei et al. investigated passive cooling strategies with a cool colored coating and PCM that increased the effectiveness throughout the whole year while reducing a cooling energy consumption by 8.5% [23]. Furthermore, Alawadhi et al. revealed that the cement roof with PCM-contained conical geometry is thermally the most effective reducing heat flux by approximately 39% [24].

In Thailand, around 28,400,000 m^2^ Autoclaved Aerated Concrete (AAC) was utilized in 2013. Its popularity was due to its lightweight, highly porous structure, higher thermal resistance, lower thermal conductivity, and faster building process than traditional concrete [25,26,27,28,29]. Of further importance, AAC shows an energy consumption saving of approximately 50%. To enhance sufficiently the thermal efficiency of AAC, the integration of phase change material (PCM) into AAC wall material was considered to reduce the propagation from the exterior surface to the interior surface. Importantly, the AAC walls with the PCM-filled conical geometry have previously not been studied. This work, therefore, concentrated on testing the further thermal performance of the AAC, by the application of various PCM-filled conical holes, to meet the best thermal performance.

## 2. Materials and Methods

### 2.1. Analysis of Behavior and Application of Phase Change Material

The thermal behavior of the paraffin PCM was examined using a differential scanning calorimeter (DSC) to investigate the material’s endothermic and exothermic peaks. Conical holes to contain the PCM were drilled into the upper surface of flat lying squares of AAC (60 cm × 60 cm × 20 cm), with the upper surface representing the future building exterior. The AAC used had a dry density of 0.61 g/cm^3^, a compressive strength of approximately 5.5 N/mm^2^, and water absorption per volume of 0.40 g/cm^3^. All values are based on Quality Class 4 of the Thai Industrial Standard (1505–1998). AAC samples were modified with 2, 3, or 4 PCM-filled conical holes (referred to as AAC-2H, -3H and -4H, respectively) in order to test each modification’s thermal performance.

### 2.2. Testing the Thermal Behavior of the AAC Modifications under Controlled Temperature Versus with Actual Weather

Figure 1 and Figure 2 show the experimental installation used to test the time lag and decrement factors of AAC, AAC-2H, AAC-3H, and AAC-4H. Each location of different thickness of original AAC, AAC-2H, AAC-3H, and AAC-4H was tested at the temperatures of 40 °C, 50 °C, and 60 °C by controlling a thermal source (heater) to observe time lag and decrement factors. This temperature range was considered as the wall temperature of most buildings in Thailand, which is in the range of 40 to 60 °C [29].

The following experiment was conducted under natural weather conditions, rather than simulated conditions. The effectiveness of the AAC was studied using four experimental rooms, as shown in Figure 3. The four test rooms were built 0.24 m^3^ in volume and the six wall sides (comprised of an area of 0.38 m^2^). One side of each testing room was constructed using various wall material types such as ordinary AAC, AAC-2H, AAC-3H, and AAC-4H. The other side of each testing room used Poly Ethylene insulation on the testing room wall. Ambient temperature, room temperature, and exterior and interior surfaces temperature were measured and recorded at 5-min intervals and continuously over 24 h. A pyranometer was set in the outdoor area to measure solar radiation intensity. Wind speed was recorded using a wind anemometer.

## 3. Results

### 3.1. Phase Change Material Behavior

The endothermic and exothermic peaks of phase change material as paraffin are shown in Figure 4. Temperature range from 0 °C to 90 °C was considered to investigate the melting and solidification point of paraffin. There are two endothermic peaks distinctively appeared in the temperature range of 0 °C to 90 °C. The first small endothermic peak was presented at approximately 41.5 °C with an involved enthalpy of 22.13 J/g, which is related to the starting state of paraffin melting. Next, there was an increase in temperature to 90 °C, a second broadening endothermic peak with an enthalpy of 137.67 J/g around 59 °C that was related to the process of the molten paraffin and continued from the first endothermic peak. Moreover, the two exothermic peaks were observed at approximately 58.5 °C and 40 °C when the temperature dropped from 90 °C to 0 °C. The first exothermic peak showed the enthalpy of 131.35 J/g because of the solidification process of some contents of paraffin. The enthalpy of 11.25 J/g was observed in the second small exothermic peak, which was related to the solidification due to heat discharge. This indicated that the condition of temperature range for the melting point, solidification point, and thermal storage capacity of paraffin PCM was suitable for the PCM integration investigation into autoclaved aerated concrete.

### 3.2. Time Lag and Decrement Factor Measurement

The wall temperature (T_w_), the time lag (ϕ), decrement factor (*f*), and the room temperature (T_R_) of AAC with different PCM-filled conical holes in each temperature condition are shown in Table 1 and Table 2. When controlling the temperature of 40 °C, the temperature in each position fluctuated, as shown in Figure 5. The exterior wall surface temperature (T_w,0_), wall temperature at the thickness of 25 mm (T_w,25_), wall temperature at the thickness of 50 mm (T_w,50_), interior wall surface temperature (T_w,75_), and room temperature (T_R_) was observed as the temperature revolution for 360 min in each condition. When the testing time was more than 180 min, the trend of wall temperature in each location was steady. The temperature value was longer than 180 min, which will be calculated to receive the average temperature in each AAC position. For the sample condition of AAC-2H, the average temperatures of T_w,0,_ T_w,25,_ T_w,50_ and T_w,75_ were around 43.3, 39.2, 35.0, and 31.7 °C, respectively (as shown in Figure 5 and Table 1). This illustrated that the greater thickness of AAC led to lower temperature. The average temperatures of other AAC samples at different locations were similar to that of the AAC-2H, as listed in Table 1.

The time lag (ϕ) and the decrement factor (*f*) [30,31] are determined by the following equations (Equations (1) and (2)).
Φ = τ_qi,max_ − τ_qe,max_(1)
(2)$$f\;=\;\frac{A_{i}}{A_{e}}\;=\;\frac{q_{i,max}\;-\;q_{i,min}}{q_{e,max}\;-\;q_{e,min}}$$
where τ_qi,max_ is the times at the maximum interior wall surface heat flux and τ_qe,max_ is the times at the maximum exterior wall surface heat flux. A_i_ is the wave amplitudes in the inner wall surface and A_e_ is the amplitudes of the wave in the outer wall surfaces. *q_i,max_, q_i,min_, q_e,max_,* and *q_e,min_* are the maximum and the minimum heat flux of the interior and exterior wall surface, respectively.

When the heat source was fixed at the temperature of 60 °C, the time lag of AAC increased from 34 to 90 min with an increase of AAC thickness from 0 to 75 mm, as shown in Table 3. For the case of AAC-2H, AAC-3H, and AAC-4H, the time lag also increased with the growth in wall thickness from 0 to 75 mm, as given in Table 3. With the thickness of 75 mm, the time lag of AAC with the 2, 3, and 4 PCM-filled conical holes is at around 90, 58, 52, and 114 min, respectively. This shows that the time lag of AAC-4H increased to around 26.7%, 96.5%, and 119.2% when compared with that of AAC, AAC-2H, and AAC-3H.

For the decrement factor of ordinary AAC at the controlled temperature of 40 °C, the value reduced from 0.328 to 0.129 with an increase of AAC wall thickness from 0 to 75 mm. In the case of controlling the temperatures of 50 °C and 60 °C, the trend of the decrement factor at the testing temperature of 50 °C and 60 °C was similar to that of the testing temperature of 40 °C when the AAC wall thickness increased from 0 to 75 mm, as illustrated in Table 2. For considering the decrement factor of AAC-2H, AAC-3H, and AAC-4H, its value decreased with higher wall thickness from 0 to 75 mm, as exhibited in Table 2. This demonstrated that an increase of AAC wall thickness is related to a reduction of the decrement factor.

The time lag of AAC-4H was the longest and the decrement factor of AAC-4H was the lowest when compared with the ordinary AAC, AAC-2H, AAC-3H, and AAC-4H in each condition. This is indicated with the AAC-4H, which can insignificantly expand the time for the heat transfer from the exterior wall surface to the interior wall surface and distinctly reduces the heat wave amplitudes. This leads to a lower room temperature for AAC-4H at approximately 1 to 3 °C when compared with that of the ordinary AAC and AAC-2H and AAC-3H, which expresses the consequence of the expanded time lag, as given in Table 4.

To investigate the thermal behaviors of the ordinary AAC, AAC-2H, AAC-3H, and AAC-4H in real weather, the four testing rooms with different conical AAC types was concurrently tested from the midnight of the day to the midnight of the following day, which gives a 24-h test cycle (October 6, October 9, October 13, and October 17–18, 2018). The evolution in solar radiation, surrounding temperature, interior wall surface temperature, exterior wall surface temperature, and room temperature evolution of the four trialing rooms were examined and compared, as illustrated in Figure 6, Figure 7, Figure 8, Figure 9, Figure 10, Figure 11, Figure 12, Figure 13 and Figure 14.

To investigate the thermal behaviors of the ordinary AAC, AAC-2H, AAC-3H, and AAC-4H in real weather, the four testing rooms with different conical AAC types was concurrently tested from midnight of one day to midnight of the following day, giving a 24-h test cycle (October 6, October 9, October 13, and October 17–18, 2018). The evolution in solar radiation, surrounding temperature, interior wall surface temperature, exterior wall surface temperature, and room temperature evolution of the four trialing rooms were examined and compared, as illustrated in Figure 6, Figure 7, Figure 8, Figure 9, Figure 10, Figure 11, Figure 12, Figure 13 and Figure 14.

The periodical cloud leads to the swinging nature of solar radiation in the tropic that is regular between sunrise (6.00 a.m.) and sunset (6.00 p.m.) with the maximum solar radiation intensity value of about 0.948 kW/m^2^ at around midday, as shown in Figure 6. Wind speed in testing the surrounding area is between 0.15 and 3.70 m/s. The surrounding temperature is affected from the weather conditions. The surrounding temperature fluctuated between ~27 °C in the morning between 5–6 a.m. and rose the maximum value by approximately 43 °C at around 3 p.m.

The temperature evolution of the ordinary AAC testing room wall is shown in Figure 7. The average maximum exterior wall surface temperature, average maximum interior wall temperature, average maximum room temperature, and average maximum ambient temperature for 5 days reached as high as approximately 50.9 °C, 49.9 °C, 43.0 °C, and 43.3 °C at around 12.00 p.m., 2.35 p.m., 3.15 p.m., and 3.00 p.m., respectively, and then decreased in value over time after each of those periods. The average exterior surface temperature, average interior wall temperature, average room temperature, and average ambient temperature throughout 5 days were at approximately 36 °C, 33.9 °C, 34.2 °C, and 32.0 °C. These temperatures were observed from midnight of one day to midnight of the following day, giving a 24-h time frame of observations in each day.

The temperature fluctuation of the AAC-2H testing room surface wall is exhibited in Figure 8. The average maximum exterior wall temperature, average maximum interior wall temperature, average maximum room temperature, and average maximum ambient temperature for 5 days reached as high as approximately 51.6 °C, 44.3 °C, 43.5 °C, and 43.3 °C at around 12.00 p.m., 2.35 p.m., 3.15 p.m., and 3.00 p.m., respectively, and then dropped in value over time passed after each those periods. The average exterior surface temperature, average interior wall temperature, average room temperature, and average ambient temperature throughout 5 days were at approximately 36.4 °C, 34.7 °C, 33.8 °C, and 32.0 °C were observed from midnight of one day to midnight of the following day, giving a 24-h time frame in each day.

The temperature evolution of the AAC-3H testing room wall is illustrated in Figure 9. The average maximum exterior wall temperature, average maximum interior wall temperature, average maximum room temperature, and average maximum ambient temperature throughout 5 days reached as high as approximately 48.2 °C, 42.5 °C, 42.6 °C, and 43.3 °C at around 12:00 p.m., 2:35 p.m., 3:15 p.m., and 3:00 p.m., respectively, and then decreased in value over time passed after each of those times. The average exterior surface temperature, average interior wall temperature, average room temperature, and average ambient temperature for 5 days were approximately 35.5 °C, 34.0 °C, 33.5 °C, and 32.0 °C, as observed from midnight of one day to midnight of the following day, giving observations for 24 h in each day.

The temperature fluctuations of the AAC-4H testing room wall is displayed in Figure 10. The average maximum exterior wall temperature, average maximum interior wall temperature, average maximum room temperature, and average maximum ambient temperature throughout 5 days reached as high as approximately 53.1 °C, 43.4 °C, 42.0 °C, and 43.3 °C at around 12:00 p.m., 2:35 p.m., 3:15 p.m., and 3:00 p.m., respectively, and then decreased in value over time passed after each of those times. The average exterior surface temperature, average interior wall temperature, average room temperature, and average ambient temperature for 5 days were at approximately 36.8 °C, 34.4 °C, 32.5 °C, and 32.0 °C were observed throughout a day from midnight of one day to midnight the following day, giving 24-h observations in each day.

Figure 11 exhibits the evolution of the exterior and interior wall surface temperature of the ordinary AAC, AAC-2H, AAC-3H, and AAC-4H in real weather. It is noted that the exterior surface temperatures of all testing rooms drastically increase from 6:00 a.m. onwards to 11:40 a.m., as presented in Figure 11a. The average maximum exterior wall temperature of the ordinary AAC, AAC-2H, AAC-3H, and AAC-4H in each day was at approximately 50.9 °C, 51.6°C, 48.2°C, and 53.1°C during the time of 12:10 to 12:30 p.m., respectively. The average maximum interior wall surface temperature of the ordinary AAC, AAC-2H, AAC-3H, and AAC-4H was at approximately 43.0 °C, 44.3 °C, 42.5 °C, and 43.4 °C at around 3 p.m. (as shown in Figure 11b), respectively.

Gradient of temperature on both the inner and outer wall surfaces of the four testing rooms was studied over 5 days, as illustrated in Figure 12. The temperature gradient on the wall surface is positive when temperature at the outer wall surface is higher than that of the interior surface, while the gradient of the temperature on the wall surface is negative when the temperature at the outer surface position is less than that of the interior surface location.

With the temperature gradient of the ordinary AAC, AAC-2H, AAC-3H, and AAC-4H testing room walls, a positive temperature gradient throughout the 5 days was first increased and reached its highest average value of around 156 °C/m, 137 °C/m, 110 °C/m, and 169 °C/m around midday and then dropped in value as time passed. The maximum negative value of the ordinary AAC, AAC-2H, AAC-3H, and AAC-4H wall was observed at about 25.2 °C/m, 32.0 °C/m, 17.7 °C/m, and 18.7 °C/m at around 6:00 p.m. This shows that the maximum positive gradient temperature of the AAC-4H wall was higher than that of an ordinary AAC, AAC-2H, and AAC-3H testing room wall, which is at around 13 °C/m, 32 °C/m, and 59 °C/m, respectively. The maximum positive temperature gradient of AAC-4H increased to around 8%, 23%, and 54% when compared with that of the ordinary AAC, AAC-2H, and AAC-3H, as shown in Figure 13. This demonstrates that the 4 PCM-filled conical holes can clearly increase the temperature difference between the exterior and interior wall surface, and between the exterior surface and room temperature, which is related to a reduction of the heat flow through the exterior wall surface to the interior wall surface. This indicates the greater insulative efficiency of AAC with the optimum PCM-contained conical hole, and implies the achievement of energy consumption saving from cooling loads in buildings. This is clearly a significant result.

Figure 14 presents the evolution of room temperature with different AAC wall patterns. The four testing rooms were observed between midnight of one day to midnight the following day. The swing nature of solar radiation, surrounding temperature, and exterior and interior surface temperature evolution affected room temperature, which varied over the day. This was seen in the room temperature of the four testing rooms, which were approximately equal between the times of 6:00 a.m. and 11:30 a.m. After that time, different peak temperatures were reached in each testing room at different times. After 11:40 a.m., the temperature readings diverged. The room temperature of AAC, AAC-2H, AAC-3H, and AAC-4H walls rose more rapidly, and achieved the highest peak value of around 43.0 °C, 43.5 °C, 42.6 °C, and 42.0 °C, at about 3:15 p.m. It was observed that the AAC-4H testing room temperature is lower than that of the AAC, AAC-2H, and AAC-3H at around 1.0 °C, 1.5 °C, and 0.6 °C, respectively. Moreover, the average room temperature throughout the 5 days of the ordinary AAC, AAC-2H, AAC-3H, and AAC-4H testing room wall is at approximately 34.2 °C, 33.8 °C, 33.5 °C, and 32.5 °C, respectively. This indicates that the use of AAC with 4 PCM-contained conical holes is the optimum condition that can significantly reduce the daily room temperature fluctuation gap. The daily room temperature swing was reduced from 15.6 °C, 15.0 °C, and 14.8 °C in the AAC room, the AAC-2H room, and the AAC-3H room to only 13.5 °C in the AAC-4H, which results from sufficient thermal absorption of the AAC-4H. This is close to the exterior surface wall. This demonstrates a decrease of heat propagation from the exterior surface to the interior surface as well as into the interior area of the testing room.

The heat flux time lag and decrement factor of the real weather conditions are determined by Equations (1) and (2). The average time lag of the original AAC, AAC-2H, AAC-3H, and AAC-4H for 5 days is at approximately 161 min, 147 min, 154 min, and 184 min and the average decrement factor of the AAC, AAC-2H, AAC-3H, and AAC-4H for 5 days is at around 0.625, 0.666, 0.731, and 0.598, respectively. This demonstrates that the samples of AAC-4H can increase the time of the transfer of the heat wave and distinctly reduces its amplitude ratio of the heat wave during this process. Considering the heat flux time lag and decrement factor comparison of AAC, AAC-2H, and AAC-3H, the 14.3%, 25.2%, and 19.5% heat flux time lag of AAC-4H was extended while the 4.3%, 10.2%, and 18.2% decrement factor of AAC-4H was reduced, respectively. This affects the room temperature of AAC-4H, which was approximately 1.7 °C, 1.3 °C, and 1 °C lower than that of the AAC, AAC-2H, and AAC-3H, which indicates the influence from an extension of the time lag, as demonstrated in Table 5. This clearly indicates that the optimum conical holes of AAC demonstrate better insulating properties, which leads to a decay of heat transmission loads. This lower room temperature and lower daily room temperature fluctuation of AAC-4H leads to obtaining energy savings in buildings and a significant decline of the yearly peak cooling requirement.

## 4. Conclusions

The thermal properties of autoclaved aerated concrete were also improved by the incorporation of phase change material. AAC with the phase change material filled four conical holes, which was the optimal condition. This can produce a time delay in the heat transfer of approximately 27%, reduce the decrement factor of around 31%, and also decreases the room temperature to around 9% when compared with conventional AAC at the controlled temperature of 60 °C. In the real weather, the AAC-4H showed thermal effectiveness by extending the time lag of approximately 14.3%, which reduces the decrement factor by approximately 4.3% and achieves approximately 5% lower room temperature when there is a comparison with the ordinary AAC. The time lag extension, decrement factor reduction, and lower room temperature resulted in the reduction of the cooling load of the testing room and more savings of electricity. This demonstrates an improvement of insulating properties corresponding to a reduction of heat wave propagation and an accomplishment of the lower room temperature, which leads to energy conservation in buildings.

## Figures and Tables

**Figure 1 materials-12-02170-f001:**
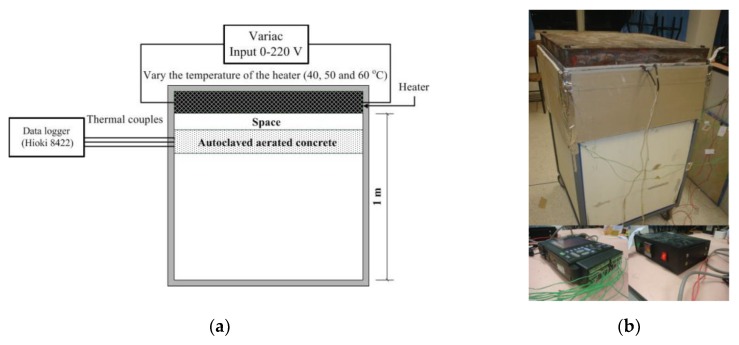
(**a**) Schematic diagram of the experimental set up. (**b**) View of the testing room and instrument installation.

**Figure 2 materials-12-02170-f002:**
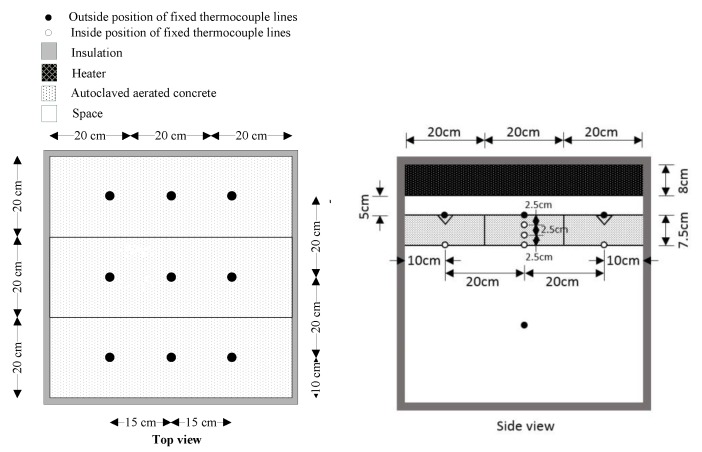
Fixed locations of thermal couples set for testing the time lag and decrement factor of samples.

**Figure 3 materials-12-02170-f003:**
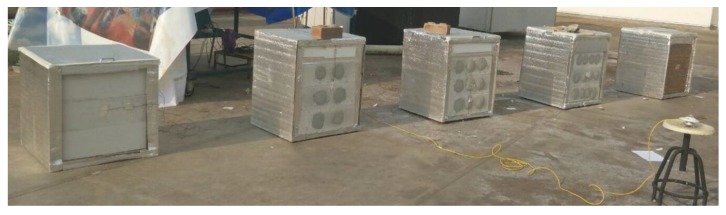
View of the testing rooms with different conical holes in real weather.

**Figure 4 materials-12-02170-f004:**
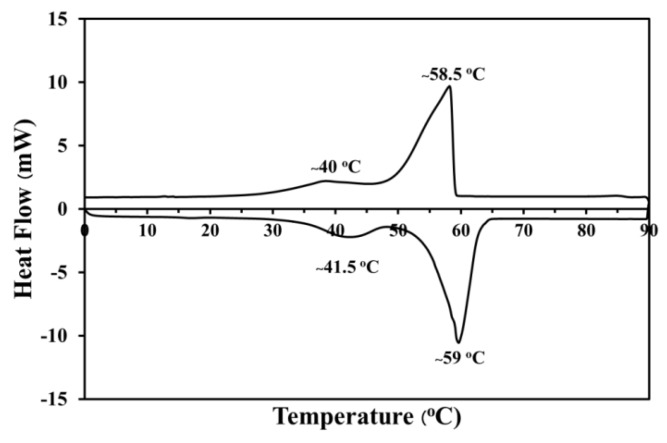
DSC plots showing the melting point of PCM.

**Figure 5 materials-12-02170-f005:**
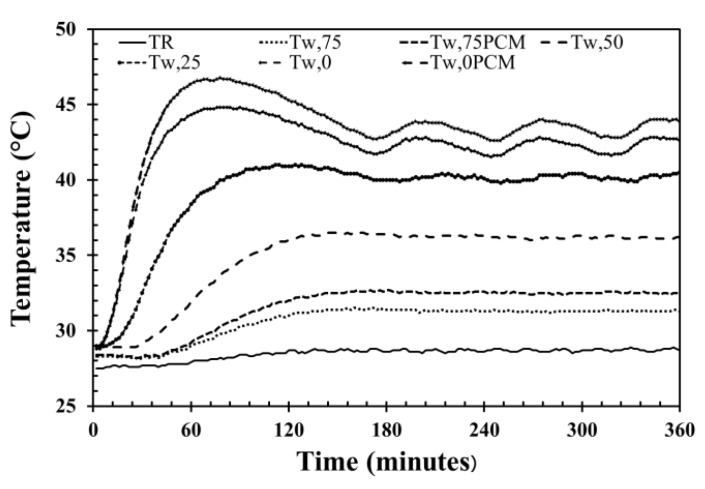
Temperature evolution of AAC wall with the two PCM-filled conical holes at 40 °C.

**Figure 6 materials-12-02170-f006:**
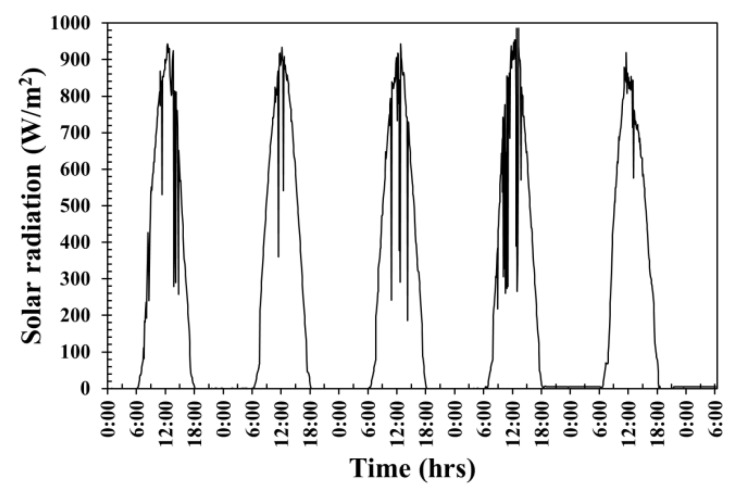
The fluctuation of solar radiation.

**Figure 7 materials-12-02170-f007:**
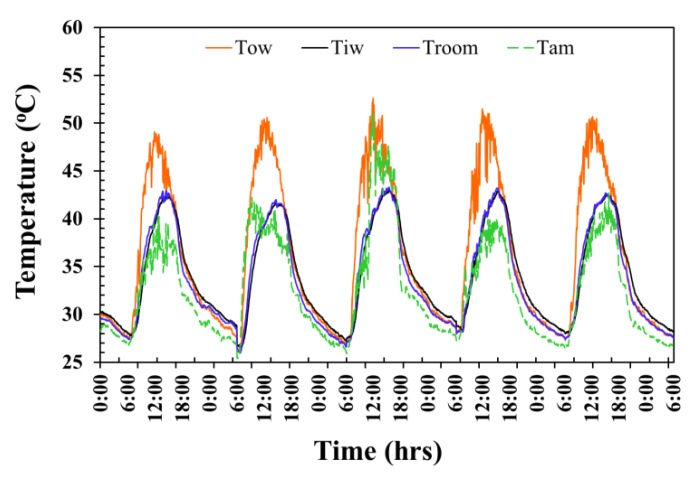
Temperature fluctuation of the ordinary AAC testing room wall.

**Figure 8 materials-12-02170-f008:**
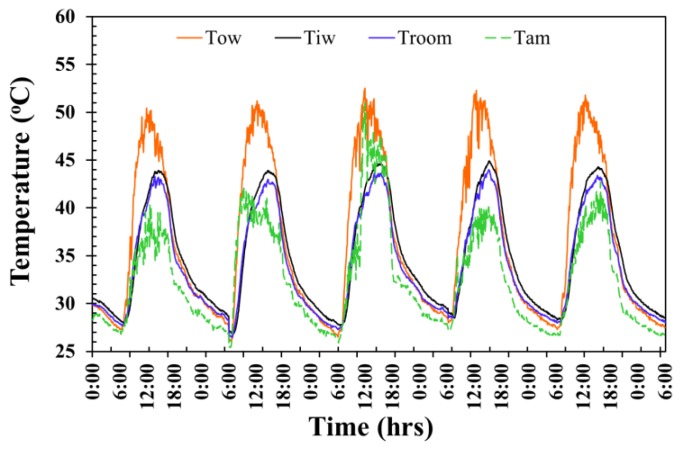
Temperature evolution of the AAC-2H testing room wall.

**Figure 9 materials-12-02170-f009:**
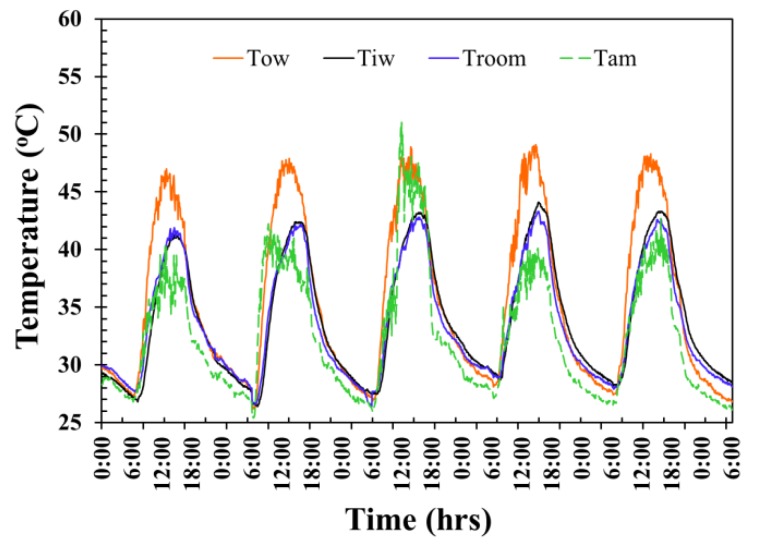
Temperature fluctuation of the AAC-3H testing room wall.

**Figure 10 materials-12-02170-f010:**
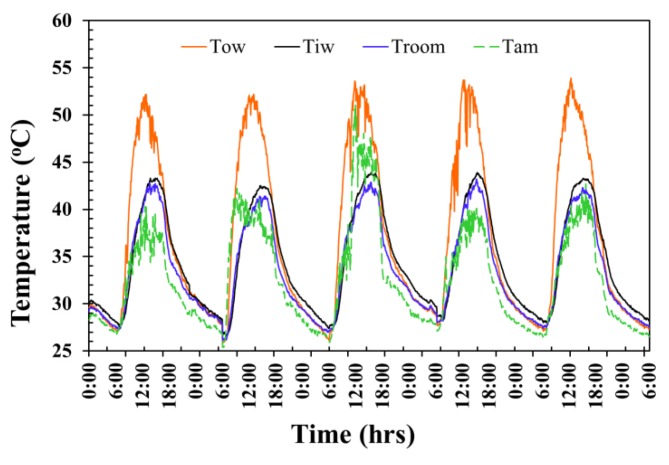
Temperature fluctuation of the AAC-4H testing room wall.

**Figure 11 materials-12-02170-f011:**
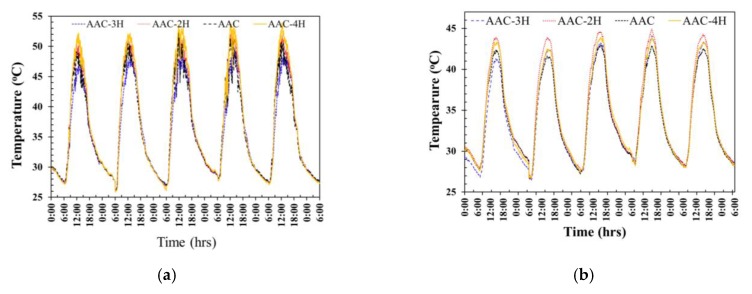
(**a**) exterior and (**b**) interior wall surface temperature of the 4 testing rooms in real weather.

**Figure 12 materials-12-02170-f012:**
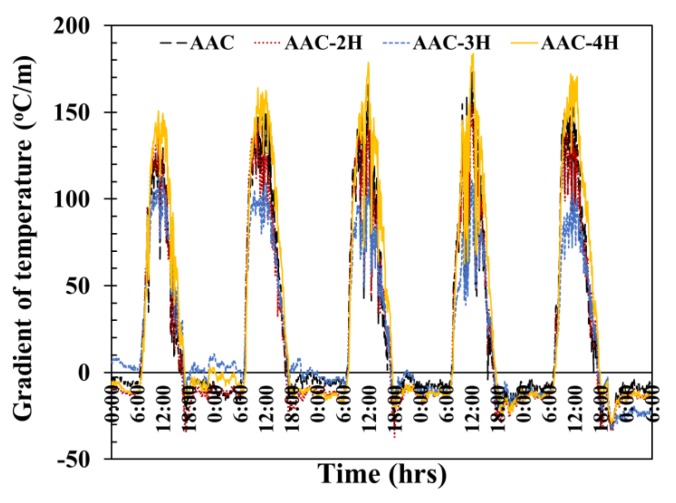
Evolution of gradient of temperature of the 4 testing rooms.

**Figure 13 materials-12-02170-f013:**
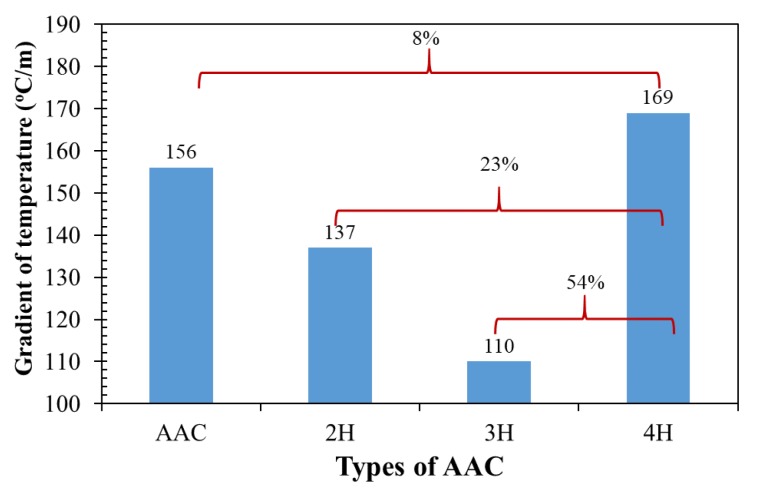
Comparison of gradient temperature of the 4 testing rooms.

**Figure 14 materials-12-02170-f014:**
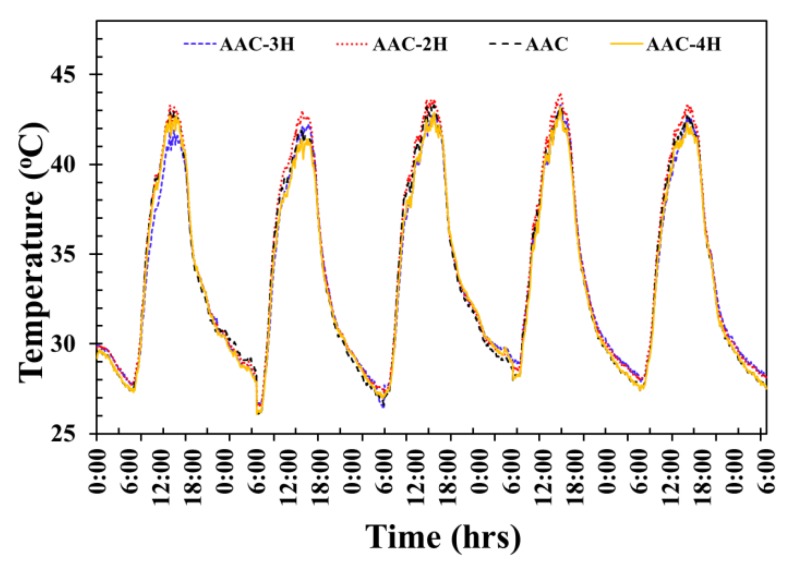
Room temperature of the 4 testing rooms in real weather.

**Table 1 materials-12-02170-t001:** Average wall temperature of AAC.

		Average Wall Temperature (°C)			
Temperature (°C)	x (mm)	AAC	AAC-2H	AAC-3H	AAC-4H
	0	43.1	43.3	42.6	42.4
40	25	38.8	39.2	38.6	38.2
	50	33.9	35.0	34.6	34.1
	75	30.9	31.7	30.9	30.8
	0	48.8	48.9	48.1	47.2
50	25	43.3	43.1	44.3	43.5
	50	36.9	37.0	39.1	37.2
	75	32.7	32.6	33.7	32.6
	0	50.5	55.7	52.5	49.2
60	25	45.5	48.6	46.7	46.2
	50	38.7	40.8	40.0	38.2
	75	33.8	34.8	33.5	31.2

**Table 2 materials-12-02170-t002:** Average decrement factor of AAC.

		Decrement Factor			
Temperature (°C)	x (mm)	AAC	AAC-2H	AAC-3H	AAC-4H
	0	0	0	0	0
40	25	0.328	0.443	0.383	0.356
	50	0.133	0.378	0.102	0.178
	75	0.129	0.195	0.054	0.089
	0	0	0	0	0
50	25	0.329	0.293	0.236	0.302
	50	0.230	0.133	0.088	0.147
	75	0.176	0.090	0.042	0.099
	0	0	0	0	0
60	25	0.302	0.326	0.418	0.254
	50	0.181	0.114	0.208	0.127
	75	0.085	0.060	0.097	0.059

**Table 3 materials-12-02170-t003:** Time lag (Φ) of AAC wall with different PCM conical holes.

Type of AAC	Time Lag (Φ) at Different Thicknesses (min)			
	0 mm	25 mm	50 mm	75 mm
AAC	0	34	78	90
AAC-2H	0	14	52	58
AAC-3H	0	16	44	52
AAC-4H	0	16	48	114

**Table 4 materials-12-02170-t004:** Average room temperature of the AAC wall with different PCM conical holes.

Average Room Temperature (°C)				
T(°C)	Ordinary AAC	AAC-2H	AAC-3H	AAC-4H
40	28.5	28.5	29.1	28.7
50	29.3	29.4	30.0	27.2
60	29.4	30.0	30.1	26.7

**Table 5 materials-12-02170-t005:** Time lag and decrement factor of the 4 testing rooms.

Material Types	Time Lag (min)						Decrement Factor					
	1st	2nd	3rd	4th	5th	Avg.	1st	2nd	3rd	4th	5th	Avg.
AAC	156	140	174	166	170	161	0.679	0.600	0.605	0.617	0.622	0.625
AAC-2H	150	110	165	160	148	147	0.688	0.665	0.660	0.661	0.653	0.666
AAC-3H	160	144	170	150	160	154	0.736	0.757	0.714	0.726	0.722	0.731
AAC-4H	170	160	220	190	180	184	0.625	0.594	0.604	0.595	0.571	0.598

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
