# Peer review of "Thermal Effectiveness Enhancement of Autoclaved Aerated Concrete Wall with PCM-Contained Conical Holes to Reduce the Cooling Load"

_materials, 2019, doi:10.3390/ma12132170_

Round 1

Reviewer 1 Report

The article entitled "Thermal effectiveness enhancement of autoclaved aerated concrete wall with PCM-contained conical holes to reduce the cooling load" presents the benefits of  using autoclaved aerated concrete (AAC) wall with phase change material (PCM). This solution fits in with the subject of modern sustainable construction and therefore this manuscript is worth publishing in Materials Journal. The subject matter of the paper is within the scope of the journal and has a good technical quality. However there are some observations and comments raised which the authors needs to address or correct:

1) During the programming of experiments two cases were accepted, i.e. 1- the controlled-temperature condition, and 2 - the real weather. Please, explain exactly the choice of these 3 temperatures (40, 50 and 60 deg.) in the first case.

2) Please provide pictures from the studies. In the curent version, only schemes of research instrumentation are shown in the article.

3) In the summary, please provide an estimate of the financial benefits of the proposed solution AAC with PCM -  used as a building material.

4) The article concerns the reduction of energy consumption, which is related to the concept of broadly understood ecology. These are original solutions and strongly promoted in construction and engineering of building materials. Therefore, in the introduction to the article one should refer to a wider range of activities undertaken in this regard. One of them is the material modification of the concrete.

This topic has already been the subject of publication in the Materials Journal and in other journals. It is therefore required that the authors comment on the results of previous papers. In the Introduction section, the following articles should be discussed and cited:

“The influence of microcrack width on the mechanical parameters in concrete with the addition of fly ash: Consideration of technological and ecological benefits”, Construction and Building Materials, 2019.

 “An assessment of microcracks in the Interfacial Transition Zone of durable concrete composites with fly ash additives”, Composite Structures, 2018.

Reviewer 2 Report

The paper concerns the investigations about the thermal performance of autoclaved aerated concrete walls with the PCM. The experiments and methods are chosen correctly to properly describe the scientific problem. The paper has a technical character, more scientific explanation of the phenomena occurring is needed. Before publication, the paper must be subjected to major revision before publication. Detailed comments are listed below:

1.      The paper should be checked by native English speaker in order to improve the style; there is also few syntax and grammar errors.

2.      Line 43-45 - The authors should discuss the results of the research to which they relate, what conclusions come from those research, etc.

3.      As the paper focuses on issues related to thermal properties and thermal storage, I suggest to give examples of other materials used in construction, which have similar properties (except for PCM materials), mainly different types of concrete. Examples of references to literature in this topic:

Pan, J., Zou, R., & Jin, F. (2017). Experimental study on specific heat of concrete at high temperatures and its influence on thermal energy storage. Energies, 10(1), 33.

Ogrodnik, P., Zegardło, B., & Szeląg, M. (2017). The use of heat-resistant concrete made with ceramic sanitary ware waste for a thermal energy storage. Applied Sciences, 7(12), 1303.

Wu, C., Pan, J., Zhong, W., & Jin, F. (2016). Testing of high thermal cycling stability of low strength concrete as a thermal energy storage material. Applied Sciences, 6(10), 271.

4.      Section 2 - please provide the basic material characteristics of the autoclaved aerated concrete that has been used.

Round 2

Reviewer 1 Report

The revised manuscript has been satisfactorily improved. Hovewer, I still have small comment to the paper. Please provide current bibliographic data of citation [7], i.e.Constr. Build. Mater. 2019, 197, 849–861

After taking this amendment into account, the article can be published.

Reviewer 2 Report

All suggestions have been included. I accept the paper for publication.
